# Efficacy of Flavonoids on Animal Models of Polycystic Ovary Syndrome: A Systematic Review and Meta-Analysis

**DOI:** 10.3390/nu14194128

**Published:** 2022-10-04

**Authors:** Jiacheng Zhang, Haolin Zhang, Xiyan Xin, Yutian Zhu, Yang Ye, Dong Li

**Affiliations:** Department of Traditional Chinese Medicine, Peking University Third Hospital, Beijing 100191, China

**Keywords:** meta-analysis, flavonoids, ovarian histomorphology, hormonal status, polycystic ovary syndrome

## Abstract

Polycystic ovary syndrome (PCOS) is one of the most common gynecological endocrinopathies. Evidence suggest that flavonoids have beneficial effects on endocrine and metabolic diseases, including PCOS. However, high-quality clinical trials are lacking. We aimed to conduct a systematic review and meta-analysis of experimental studies to determine the flavonoids’ effects in animal models of PCOS. Three electronic databases including PubMed, Scopus, and Web of Science were systematically searched from their inception to March 2022. The Systematic Review Center for Laboratory Animal Experimentation’s risk of bias tool was used to assess methodological quality. The standardized mean difference was calculated with 95% confidence intervals as the overall effects. R was used for all statistical analyses. This study was registered in PROSPERO (registration number: CRD42022328355). A total of eighteen studies, including 300 animals, met the inclusion criteria. Our analyses demonstrated that, compared to control groups, flavonoid groups showed a significantly lower count of atretic follicles and cystic follicles and the count of corpus luteum was higher. A significant reduction in the luteinizing hormone (LH), LH/follicle-stimulating hormone (FSH), and free testosterone were observed in intervention groups. Nevertheless, there was no significant difference in the effects of flavonoids on the level of FSH, estradiol, and progesterone. Subgroup analyses indicated that the type of flavonoid, dose, duration of administration, and PCOS induction drug were relevant factors that influenced the effects of intervention. Current evidence supports the positive properties of flavonoids on ovarian histomorphology and hormonal status in animal models of PCOS. These data call for more randomized controlled trials and further experimental studies investigating the mechanism in more depth.

## 1. Introduction

Polycystic ovary syndrome (PCOS), one of the most common gynecological endocrinopathy, affecting approximately 8% to 18% women in the premenopausal period [1]. According to the Rotterdam conference [2], PCOS is mainly characterized by androgen excess and ovarian dysfunction (oligo-anovulation or polycystic ovarian morphology). The pathological mechanism of PCOS is generally associated with the hypothalamic–pituitary–ovarian axis [3]. In PCOS, hypothalamic gonadotropin-releasing hormone (GnRH) pulses are activated, and the release of the luteinizing hormone (LH) is enhanced relative to the follicle stimulating hormone (FSH), which makes theca cells preferentially secrete more androgens [4]. Hyperandrogenism affects follicular development via a complex mechanism, such as insulin resistance and dyslipidemia. It is difficult to distinguish the causal relationship between these pathological factors, as they jointly form a vicious cycle of aggravating PCOS [5]. Additionally, various studies have indicated that systemic low-grade inflammation and oxidative stress are associated with the development of PCOS [6,7].

Despite the substantial burden PCOS has caused, no drug has been approved specifically for it by the Food and Drug Administration nor the European Medicines Agency [8,9]. Current medication treatments for PCOS, including letrozole, oral contraceptives, anti-androgens, and clomiphene are suboptimal. The side effects of these drugs, including clinical resistance and nausea, along with ovarian hyperstimulation syndrome, are prone to occur with long-term treatments [10,11,12]. Non-invasive, symptom-oriented, and preventative treatments are sorely needed.

The first-line therapy for PCOS patients with mild symptoms is lifestyle modification [13,14]. Dietary intervention is a promising strategy for the treatment of PCOS that has been widely recognized [15,16]. Among the phytonutrients in dietary macronutrients, flavonoids have attracted considerable attention for their potential antioxidant and free-radical scavenging effects against metabolism and endocrine-related diseases [17,18]. The basic structure of flavonoids includes the common C6C3C6 skeleton and consists of two phenyl rings and one oxygenated heterocyclic ring (Appendix A) [19]. Based upon variations in the heterocyclic ring, flavonoids can be divided into anthocyanidins, flavanols (flavan-3-ols or catechins), flavanones, flavones, flavonols, and isoflavones [20]. Accumulating studies suggest that the bioavailability of flavonoids is higher than previously thought [21]. In recent years, clinical trials using specific flavonoids to manage PCOS have been carried out on a small scale. Previous studies explored the efficacy of soy isoflavones in PCOS patients. The results indicated that after the 12-week intervention of 50 mg/day soy isoflavones, markers of insulin resistance, hormonal status, lipid profiles, and oxidative stress were partly alleviated in women with PCOS [22]. Another study assessed the therapeutic effects of puerarin in patients with PCOS. Compared with before the treatment, significantly improved levels of sex hormone binding globulin and superoxide dismutase were observed [23]. A systematic review including five experimental studies and three clinical trials showed the beneficial effects of quercetin on ovarian histomorphology, hormone disorders, and dyslipidemia, while no significant effect was reported for weight loss [24]. To date, there is still a lack of rigorous clinical trials to investigate the efficacy of flavonoids on PCOS patients. Among the published randomized controlled trials, a small number of biases related to health status, genetic background, or methodology have been observed [24]. Reviews based on preclinical studies may assist in offering solid evidence and informing future experimental and clinical trials. Additionally, mice and rats are ideal animal models for PCOS because they are sensitive to hormone stimulation and possess a stable estrous cycle [25].

Herein, we report on a systematic review and meta-analysis of data from studies testing the efficacy of flavonoids on animal models of PCOS. The changes in ovarian histomorphology and hormonal status were included as observation parameters. In addition, we evaluated whether the effects differ in terms of the type of flavonoid, dose, treatment duration, and PCOS induction drug by subgroup analysis.

## 2. Methods

This systematic review was conducted in accordance with the Preferred Reporting Items for Systematic Reviews and Meta-Analysis guidelines [26]. The protocol based on SYRCLE’s tool for animal studies [27] was registered in PROSPERO (registration number: CRD42022328355).

### 2.1. Search Strategy

We conducted a systematic search of PubMed, Web of Science, and Scopus from inception to March 2022. The language of publications was limited to English. The specific search items included (“anthocyanidins” OR “flavanols” OR “flavan-3-ols” OR “catechins” OR “flavanones” OR “flavones” OR “flavonols” OR “isoflavones”) AND (“polycystic ovary syndrome” OR “polycystic ovarian syndrome” OR “PCOS”) AND (“mice” OR “mouse” OR “rat” OR “rats” OR “animal”). Additionally, a manual search was conducted to check the relevant publications by two authors.

### 2.2. Inclusion and Exclusion Criteria

Studies were considered eligible based on the following inclusion criteria: (1) The participants were animal models of PCOS; (2) the intervention drugs were flavonoids (pure flavonoids or flavonoids extracts) and the type of flavonoid and dose were clarified; (3) the comparison was the PCOS induction group with no treatment; (4) the outcomes included the effects of flavonoids on the development of PCOS, histomorphology, and hormonal alternations in animal models, and the original trials should report one or more following outcomes: the count of atretic follicles, the count of cystic follicles, the count of corpus luteum, LH, FSH, LH/FSH, free testosterone (FT), estradiol, and progesterone; and (5) studies used animal models and were published in English.

Two reviewers examined the titles and abstracts of retrieved studies. The exclusion criteria were as follows: (1) Non-original full research articles; (2) clinical trials, in vitro models, retrospective studies, case reports, and protocols; (3) interventions different from flavonoids or without precise dose and duration of administration; and (4) the presence of concomitant interventions in the PCOS group. Further, full texts were assessed by two reviewers, and publications without relevant outcomes were excluded.

### 2.3. Dara Extraction

Two reviewers independently assessed the extraction of data from selected literature. Any difference was resolved by discussion with the third reviewer. The following essential details were summarized as the baseline characteristics of the studies: (1) Publication details (author and year); (2) intervention performed (type of flavonoid, dose, route, and duration of administration); (3) PCOS induction drug and methods; (4) animal used (species, strain, age, and weight); (5) outcomes included.

All the data of outcome measures were continuous. We extracted data reporting the sample size per group (*N*), mean values, and variance [standard deviation (SD) or standard error of mean (SEM)]. SEM was converted to SD by using the formula ($SD=SEM\times \sqrt{N}$). When treatment was administrated in multiple doses, the group using the highest dose was recorded [28]. In case the outcomes were only presented graphically, the reviewers used the ImageJ software to quantify the results.

### 2.4. Quality Assessment

Two reviewers independently assessed the internal validity of the included publications, referencing the SYCLE’s risk of bias tool for animal experiments [27]. This six-part checklist of evaluation included: (1) Selection bias (sequence generation, baseline characteristics, and allocation concealment); (2) performance bias (random housing and blinding of trial caregivers); (3) detection bias (random outcome assessment and blinding of outcome assessors); (4) attrition bias (incomplete outcome data); (5) reporting bias (selective outcome reporting); and (6) other bias (assessment of PCOS model, temperature control, drug production institutions, conflict of interest, et al.). Any discrepancy was discussed with the third reviewer.

### 2.5. Statistical Analysis

R software (V4.1.3) was adopted for data analysis and visualization (package meta and dmetar). All the outcome measures were continuous, and the standardized mean difference (SMD) was calculated with 95% confidence intervals (CIs) as the overall effects. A random-effect model was performed. Heterogeneity was assessed by the Q statistic and quantified using the $I^{2}$ results [29]. p<0.05 was considered statistically significant. When $I^{2}>50\%$, subgroup analyses were conducted to explore the sources of heterogeneity. The type of flavonoid, dose, duration of administration, and PCOS induction drug were considered as the potential subgroup basis. Sensitivity analyses were performed to confirm the robustness of the results by removing one study and repeating the meta-analysis. Publication bias was assessed with the trim-and-fill method, and the Egger’s bias test was performed if the results contained at least ten studies.

## 3. Results

### 3.1. Study Selection

Initially, we retrieved 327 studies through a comprehensive search, out of which 181 non-duplicate publications were filtered out. Based on the predetermined exclusion criteria, 156 studies were removed. After the full text assessment, 7 studies were excluded, and 18 eligible publications were included in this systematic review. A flowchart depicting the process of selection is presented in Figure 1.

### 3.2. Study Characteristics

A total of 18 articles [30,31,32,33,34,35,36,37,38,39,40,41,42,43,44,45,46,47] investigated 7 types of flavonoids on PCOS animal models. All the included studies were conducted between 2015 and 2022. Only 2 studies used mice, and rats were used in the remaining 16. In 10 out of these 16 studies, Sprague Dawley rats were incorporated, and Wistar rats were performed in the remaining 6 publications. The flavonoids administration dose varied greatly, with it ranging from 20 mg/kg/day to 200 mg/kg/day. The duration of administration ranged from two weeks to six weeks. The PCOS models were induced by letrozole (in seven studies), dehydroepiandrosterone (DHEA) (in six studies), estradiol valerate (in two studies), testosterone propionate (TP) (in one study), testosterone enanthate (TE) (in one study), and insulin combined with human chorionic gonadotropin (hCG) (in one study). With regard to the outcomes of interest to us in the 18 studies, 5 comparisons for the count of atretic follicles, 8 comparisons for the count of cystic follicles, and 9 comparisons for the count of corpus luteum were the primary outcome measurements. LH (in 11 comparisons), FSH (in 11 comparisons), LH/FSH (in 8 comparisons), free testosterone (FT) (in 16 comparisons), estradiol (in 12 comparisons), and progesterone (in 9 comparisons) were assessed as the secondary outcome measurements. The detailed characteristics were summarized in Table 1.

### 3.3. Study Quality

The methodological quality assessment is shown in Figure 2, and the detailed information of each study is provided in Appendix A. None of the studies mentioned the following aspects: Allocation concealment, the blinding of trial caregivers, random housing, and random outcome assessment. The selection bias and detection bias of most studies were unclear (77.8% and 88.9%, respectively). A total of 16 studies showed a low risk of attrition bias (88.9%), and 15 studies showed a low risk of reporting bias (83.3%). The evaluation of PCOS model establishment, comorbidity, temperature control, and drug production institutions were considered in other bias. Three studies showed a high risk of other bias (16.7%), nine studies showed a low risk of other bias (50%), and the remaining six were unclear (33.3%). Although the general quality of the publications was not satisfactory, no literature was excluded for its quality.

### 3.4. Primary Outcomes

#### 3.4.1. Count of Atretic Follicles

Five comparisons measured the influence of flavonoids on the count of atretic follicles. The pooled effects showed that the administration of flavonoids was associated with a significant difference compared with the control group (SMD = −1.73, 95%CI: −2.30 to −1.16). There was no significant heterogeneity among these studies ($I^{2}=0\%,\;p=0.86)$ (Figure 3).

#### 3.4.2. Count of Cystic Follicles

Eight comparisons reported the count of cystic follicles. The pooled results showed that the administration of flavonoids was associated with a significant difference compared with the control group (SMD = −3.36, 95%CI: −5.36 to −1.36). Heterogeneity was considerable ($I^{2}=87\%,\;p<0.01)$ (Figure 4).

#### 3.4.3. Count of Corpus Luteum

Nine comparisons reported the count of corpus luteum. The pooled results showed that the administration of flavonoids was associated with a significant difference compared with the control group (SMD = 2.41, 95%CI: 1.12 to 3.70). Heterogeneity was considerable ($I^{2}=83\%,\;p<0.01)$ (Figure 5).

### 3.5. Secondary Outcomes

#### 3.5.1. LH

Twelve comparisons reported LH. The pooled effects show the administration of flavonoids was associated with a significant difference compared with the control group (SMD = −2.92, 95%CI: −4.02 to −1.82). Heterogeneity was considerable ($I^{2}=85\%,\;p<0.01)$ (Figure 6).

#### 3.5.2. FSH

Eleven comparisons reported FSH. The pooled effects showed that there was no significant difference in the effects of flavonoids on the level of FSH (SMD = 0.39, 95%CI: −1.02 to 1.81). Heterogeneity was considerable ($I^{2}=88\%,\;p<0.01)$ (Appendix A).

#### 3.5.3. LH/FSH

Eight comparisons reported LH/FSH. The pooled effects showed that the administration of flavonoids was associated with a significant difference compared with the control group (SMD = −3.01, 95%CI: −4.81 to −1.20). Heterogeneity was considerable ($I^{2}=77\%,\;p<0.01)$ (Figure 7).

#### 3.5.4. FT

Sixteen comparisons reported FT. The pooled effects showed that the administration of flavonoids was associated with a significant difference compared with the control group (SMD = −3.54, 95%CI: −5.10 to −1.99). Heterogeneity was considerable ($I^{2}=86\%,\;p<0.01)$ (Figure 8).

#### 3.5.5. Estradiol

Fifteen comparisons reported estradiol. The pooled effects showed that there was no significant difference in the effects of flavonoids on the level of estradiol (SMD = 1.42, −0.47 to 3.32). Heterogeneity was considerable ($I^{2}=91\%,\;p<0.01)$ (Appendix A).

#### 3.5.6. Progesterone

Eight comparisons reported progesterone. The pooled effects showed that there was no significant difference in the effects of flavonoids on the level of progesterone (SMD = 0.46. −3.49 to 4.41). Heterogeneity was considerable ($I^{2}=94\%,\;p<0.01)$ (Appendix A).

### 3.6. Subgroup Analysis

The pooled estimates for studies in the meta-analysis of the count of cystic follicles, the count of corpus luteum, LH, LH/FSH, and FT exhibited substantial heterogeneity. Subgroup analyses used to explore the sources of heterogeneity were identified with four covariates (type of flavonoid, dose, duration of administration, and PCOS induction drug). The flavonoids dose was divided into three groups: low (≤50 mg/kg/day), medium (>50 mg/kg/day and ≤100 mg/kg/day), and high (>100 mg/kg/day). Additionally, the duration of administration was divided into three groups: short (≤2 weeks), medium (>2 weeks and ≤4 weeks), and long (>4 weeks).

For the count of cystic follicles, we found that the type of flavonoid may be the possible source of heterogeneity (p<0.01). Only the study where animals administrated with soy isoflavone showed no significant difference in the effects of flavonoids (SMD = 0.12, 95%CI: −0.86 to 1.10). We did not find any statistical difference in subgroups based on dose, duration of administration, and PCOS induction drug. Considering each subgroup separately, we found the count of cystic follicles did not decrease significantly in studies using letrozole as the induction drug (SMD = −3.57, 95%CI: −9.21 to 2.07) (Table 2 and Appendix A). For the count of corpus luteum, we found that the sources of heterogeneity may be the type of flavonoid (p=0.03) and PCOS induction drug (p=0.02). No significant difference was shown in studies whereas animals administrated with soy isoflavone (SMD = 0.82, 95%CI: −0.22 to 1.85) or anthocyanin (SMD = 0.87, 95%CI: −0.06 to 1.61). Similar results were obtained in studies whose PCOS induction drug was TE (SMD = 0.78, 95%CI: −0.06 to 1.61) or TP (SMD = 0.30, 95%CI: −0.51 to 1.11) (Table 3 and Appendix A).

Meanwhile, we explored the sources of heterogeneity in hormonal status studies. We found that dose (p<0.01), duration of administration (p=0.03), and PCOS induction drug (p=0.03) were the factors that interfered with the effect of flavonoids on LH levels in the PCOS model. The pooled estimates showed significant differences between a high dose (SMD = −5.17, 95%CI: −6.54 to −3.80), a medium dose (SMD = −2.52, 95%CI: −3.13 to −1.90), and a low dose (SMD = −2.40, 95%CI: −4.63 to −0.17). We did not find statistical differences between studies where animals were administrated with various types of flavonoids (Table 4 and Appendix A). For LH/FSH, no significant differences were observed through subgroups of type of flavonoid (p=0.3), dose (p=0.08), duration of administration (p=0.17), and PCOS induction drug (p=0.4) (Table 5 and Appendix A). For FT, we found that the type of flavonoid may be the source of heterogeneity (p=0.03). No significant difference was shown in studies where animals administrated were with rutin (SMD = −0.56, 95%CI: −1.84 to 0.71) or baicalin (SMD = −5.3, 95%CI: −14.06 to 3.46) (Table 6 and Appendix A).

### 3.7. Sensitivity Analysis

For the count of atretic follicles, the count of cystic follicles, the count of corpus luteum, LH, LH/FSH, and FT, a sensitivity analysis was performed to confirm and account for the stability of the positive results by the leave-one-out test. Overall, the pooled effects did not change significantly in these six cases, which suggested the results remained robust (Figure 9).

### 3.8. Publication Bias

The Egger’s bias test showed that, except for the count of atretic follicles (p=0.66), substantial publication bias was detected for the remaining outcomes (p<0.01, both) (Appendix A). A trim-and-fill evaluation was then conducted, and substantial asymmetry is indicated in Appendix A. Generally speaking, the publication bias of the results should not be ignored.

## 4. Discussion

Flavonoids are widely distributed plant secondary metabolites which are found in various fruit, vegetables, and herbal medicines. They play important roles in human health through the consumption of plant-derived foods by scavenging free radicals and inhibiting metal-ion chelators for their powerful antioxidant properties [48]. Flavonoids are also associated with the modulation of the immune system, despite that they represent ancillary ingredients with immunomodulatory properties that require more evidence [49]. Nevertheless, accumulating evidence demonstrates that flavonoids can inhibit regulatory enzymes and transcription factors involved in inflammation [50,51]. On account of their diverse bioavailability, flavonoids have been applied to prevent degenerative diseases, such as diabetes, cardiovascular complications, cancer, and hypoglycemia [52,53]. Additionally, significant applications of flavonoids have been unveiled in reproductive endocrine diseases, such as menopausal syndrome and endometriosis [54,55]. PCOS is an inflammatory, systematic, and autoimmune endocrinopathy [56]. In PCOS patients, systematic low-grade inflammation compromises multiple aspects of fertility and is associated with hyperandrogenism and insulin resistance [57]. Therefore, we speculate that flavonoids can ameliorate symptoms of PCOS. Our results demonstrate the efficacy of flavonoids on PCOS animal models.

The ovarian follicles of PCOS patients are manifested with a thickened theca cell layer and cyst formation. Most studies purport that histological changes such as follicular atresia are the cause of PCOS infertility [58]. Inconsistent with previous PCOS-related meta-analysis which were based on clinical trials, histopathological changes were taken as the primary outcomes in our study. The convenient collection and observation of ovarian tissue is a major advantage of preclinical studies over clinical trials. Although the results of histomorphology were statistically positive, substantial heterogeneity was detected regarding the count of cystic follicles and the count of corpus luteum. Results from the subgroup analyses suggest that only one study which used soy isoflavone did not show a statistic reduction of the count of cystic follicles. In this study, the treatments were divided into three groups: the soy isoflavone group, the resistant starch group, and the soy isoflavone combined with resistant starch group, which individually showed a significant reduction in cyst formation [31]. We only included the soy isoflavone group as the intervention method, which may have led to the heterogeneity. Similar results were obtained in measurements of the count of corpus luteum. In the letrozole-induced rats, a larger number of cystic follicles and a smaller number of corpus luteum were observed. Abnormalities in terms of follicle development occurred not only in the later, antral stages of follicles which are gonadotrophin dependent but also in the very earliest stages of folliculogenesis [59]. The changes were related to LH and FSH disorders and a lack of interplay between granulosa cells [60]. Soy isoflavones modulated hormone levels by binding to estrogen receptors, and the property may be enhanced by butyric acid, which was elevated by resistant starch intake [61]. Additionally, the studies which applied TE and TP as PCOS induction drugs did not show significant improvements in terms of the corpus luteum count. A systematic review of PCOS animal models demonstrated that hormonal interventions using androgens promote the most consistent features of PCOS morphological phenotypes [62]. Contrary to our expectations, the dose and duration of treatments were irrelevant to the efficacy of flavonoids, which revealed that the morphological changes of PCOS models are relatively fixed.

In addition, most of the included studies indicated a role for flavonoids in modulating hormonal status. The increased levels of LH/FSH and testosterone were due to the impaired hypothalamic–pituitary axis [63]. LH is a central actor in theca cell dysregulation, which follows ovarian hyperandrogenism. Previous studies demonstrated that estrogen stimulated by LH is beneficial to the maturation of oocyte cytoplasm and membrane, which revealed the importance of LH [64]. Additionally, in the middle and later stage of follicle development, granulosa cells begin to express luteinizing hormone receptor (LHCGR), and LH reaches the peak [65]. Meanwhile, LH and FSH cooperate to stimulate ovulation and promote granulosa cell luteinization [66]. Meanwhile, in PCOS patients, the expression of LHCGR is premature in granulosa cells [67]. Continuous estrogen enhances the sensitivity of the pituitary gland to GnRH secreted by the hypothalamus, which increases the frequency and amplitude of GnRH pulse secretion and increases the level of LH. Because the negative feedback of hormones on FSH is greater than that on LH, the ratio of LH to FSH is higher [68,69]. Our meta-analysis revealed the downregulation of LH, LH/FSH, and FT with the administration of flavonoids. The dose, duration of administration, and PCOS induction drug were the main sources of heterogeneity in LH reduction. As we expected, with higher dose of flavonoids, there were lower levels of LH. Only the studies which applied letrozole as the induction drug did not show significant a reduction in LH. As for FT, the effects were not statistically different in studies using baicalin or rutin as the treatments. However, we did not find the sources of heterogeneity in the LH/FSH studies. Considering each study separately, we found the studies which used a medium dose showed better a reduction in terms of LH/FSH than those which used a low dose.

Among the studies included, PCOS induction drugs were divided into androgens (TP, TE, and DHEA), estrogens, aromatase inhibitors (letrozole), and insulin combined with hCG. Androgen induction may promote continuous high blood-free testosterone and a pathologic elevation in FSH that induces cystic formation [70,71]. High estrogen stimulation leads to the degeneration of hypothalamic neurons and the compensatory hyperplasia of the pituitary gland, which increases the sensitivity of the pituitary gland to GnRH. The level of LH increases and the secretion of FSH is inhibited, resulting in the typical characteristics of PCOS [72]. Letrozole, as a nonsteroidal aromatase inhibitor, restrains the conversion of androgen to estrogen, leading to androgen accumulation [73]. Insulin combined with hCG may destroy the normal pulse secretion mode of endogenous LH and is characterized by hyperandrogenism and insulin resistance [74]. As PCOS is a highly heterogeneous disease, a model that fully simulates the characteristics of PCOS does not exist. The appropriate modeling methods should be applied according to the aim of the study.

To date, there have been only a small number of meta-analyses of PCOS in animal models. This might be explained by doubts about the substantial heterogeneity caused by the diversity of modeling methods and applied drugs. In order to better take advantage of the systematic review of preclinical studies and explain the changes induced by flavonoids on different parameters of PCOS, it is important to consider the molecular mechanisms by which flavonoids may produce effects. The literature reports that soy isoflavones were demonstrated to enhance the antioxidant capacity of rats and inhibit the activation of the nuclear factor-kappa beta (NF-κB) signaling pathway, hence reducing inflammatory cytokines [32]. Similar results were reported in studies using catechins as treatments, with NF-κB-mediated inflammation and matrix metallopeptidase 2 and matrix metallopeptidase 9-mediated damage being ameliorated. Additionally, the signal transducer and activator of transcription 3 signaling was inhibited [35]. Quercetin, a flavonol, was reported to decrease the expression of the CYP17A1 gene by inhibiting phosphatidylinositol 3-kinase (PI3K), leading to the regulation of ovarian steroidogenesis [37]. Furthermore, another study suggested that quercetin has an apoptosis-inhibiting effect through increasing B-cell lymphoma-2 (Bcl2) and decreasing the Bcl2-Associated X (BAX) to Bcl2 ratio [38]. The ability to restore the maturation of oocyte and regulate energy homeostasis was emphasized after the administration of quercetin [39,40]. The literature reports that the upregulation of adenosine 5′-monophosphate-activated kinase and activation of PI3K signaling contributed to the beneficial effect of baicalin on PCOS [42]. Another publication revealed that GATA1 is one of the key genes affected by baicalin. PCOS models reversed the hyperandrogenic status after baicalin treatments [43]. Rutin has metformin-like properties which play an important role in reducing reactive oxygen species and boosting the antioxidant status [44]. The potential molecular mechanisms of various flavonoid effects on PCOS are illustrated in Appendix A.

Our study also has many limitations. First, the language of publications was limited in English, so databases in other languages were excluded. The gray literature and negative results were also relatively lacking. Second, few studies measured pregnancy outcomes, which means the effect of flavonoids on infertility in animal models could not be assessed. Furthermore, metabolic disorders and insulin resistance were not evaluated. Third, although some aspects of heterogeneity were explained by different experimental designs, the remaining heterogeneity and publication bias should be valued. Finally, considering the great difference between species, the ovulation characteristics of humans and rodents should be considered. When our results are referenced by clinical protocols, the standard dose conversion should be conducted.

## 5. Conclusions

To the best of our knowledge, this is the first study aimed at systematically reviewing the efficacy that flavonoids have on the pathology of PCOS, especially on ovarian histomorphology and hormonal status. It also analyzed how different types of flavonoids influenced various phenotypes in animal models induced by different drugs. Specifically, the count of atretic follicles, the count of cystic follicles, LH, LH/FSH, and FT reduced, and the count of corpus luteum increased in the groups where animals were administrated with flavonoids. There was no statistical difference in comparisons of FSH, estradiol, and progesterone. Furthermore, appropriate PCOS modeling methods meeting various mechanisms should be taken into consideration. With regard to the substantial heterogeneity and publication bias, the results must be interpreted with prudence.

## Figures and Tables

**Figure 1 nutrients-14-04128-f001:**
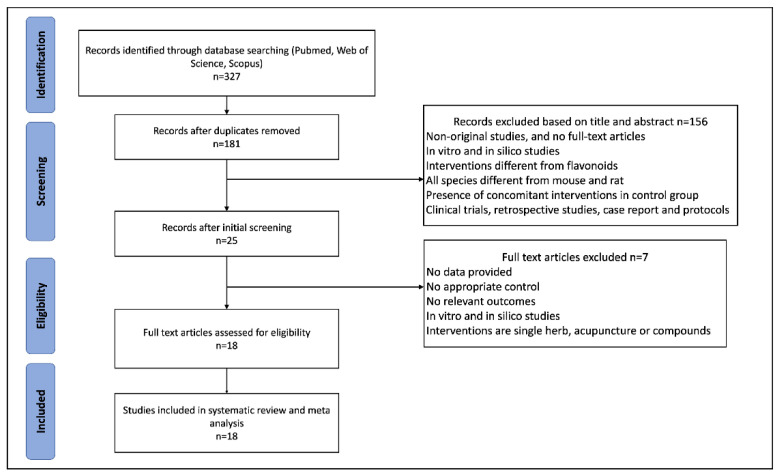
Flowchart of the study selection process.

**Figure 2 nutrients-14-04128-f002:**
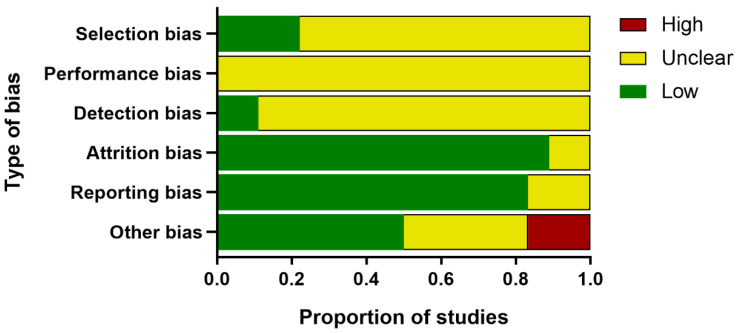
Study quality assessed through SYRCLE’s risk of bias tool for animal studies.

**Figure 3 nutrients-14-04128-f003:**
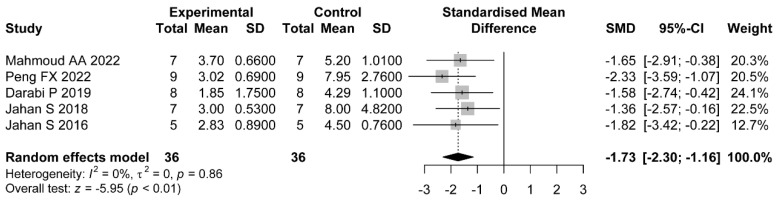
Forest plot of flavonoids for the count of atretic follicles [36,37,38,44,46,47].

**Figure 4 nutrients-14-04128-f004:**
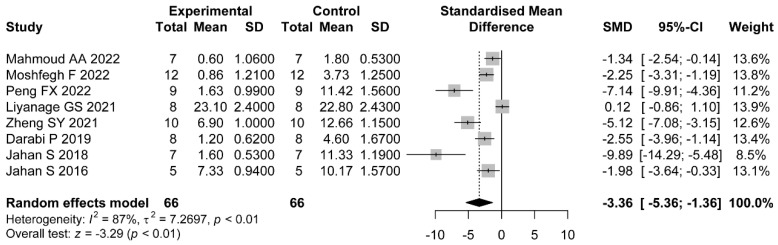
Forest plot of flavonoids for the count of cystic follicles [31,36,38,39,41,44,46,47].

**Figure 5 nutrients-14-04128-f005:**
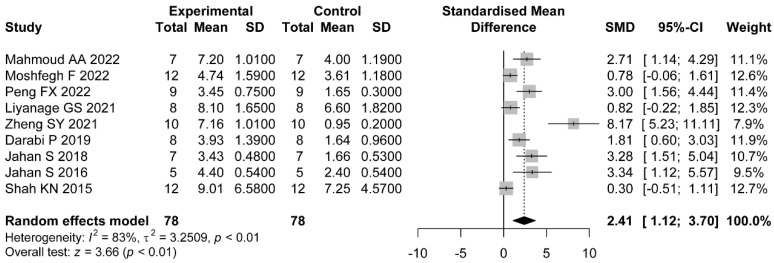
Forest plot of flavonoids for the count of corpus luteum [31,36,37,38,39,41,44,46,47].

**Figure 6 nutrients-14-04128-f006:**
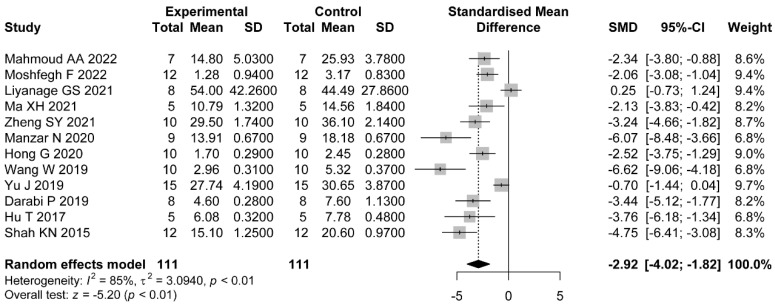
Forest plot of flavonoids for LH [31,32,33,35,37,38,41,42,43,45].

**Figure 7 nutrients-14-04128-f007:**
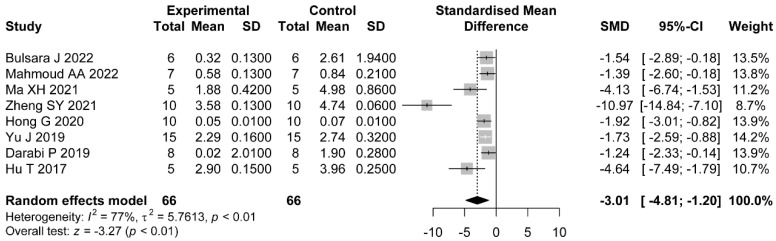
Forest plot of flavonoids for LH/FSH [30,32,35,38,39,43,45,46].

**Figure 8 nutrients-14-04128-f008:**
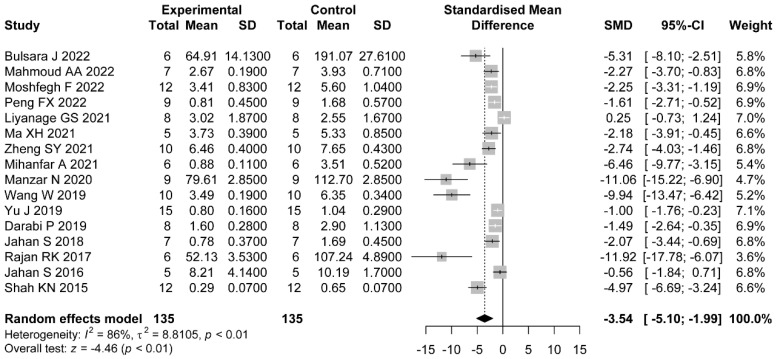
Forest plot of flavonoids of FT [30,31,32,33,34,36,37,38,39,40,41,42,43,44,46,47].

**Figure 9 nutrients-14-04128-f009:**
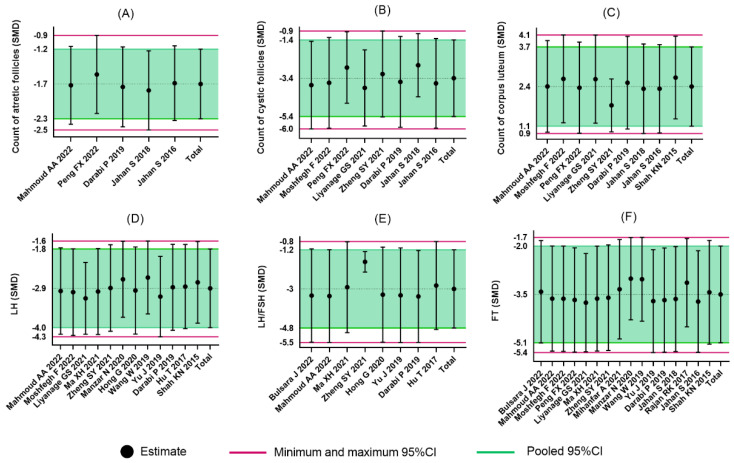
Sensitivity analyses of (**A**) the count of atretic follicles, (**B**) the count of cystic follicles, (**C**) the count of corpus luteum, (**D**) LH, (**E**) LH/FSH, and (**F**) FT.

**Table 1 nutrients-14-04128-t001:** Characteristics of 18 included studies.

Author, Year	Type of Flavonoids	Flavonoid Subclass	Route	Dose	Duration	PCOS Induction Drug	PCOS Induction Method	Species	Strain	Outcomes
Bulsara, J., 2022 [30]	Soy isoflavone	Isoflavones	oral	100 mg/kg/d	2 weeks	Letrozole	Letrozole p.o. at 1 mg/kg dissolved in 0.5% CMC daily for 21 days.	Rat	SD	⑥⑦⑧⑨
Mahmoud, A.A., 2022 [38]	Quercetin	Flavonols	oral	25 mg/kg/d	4 weeks	DHEA	DHEA at 60 mg/kg per 1 mL sesame oil for 41 days.	Rat	Wistar	①②③④⑤⑥⑦⑧
Moshfegh, F., 2022 [41]	Anthocyanin	Anthocyanins	injection	80 mg/kg/d	2 weeks	TE	TE s.c. in the back of the neck (1 mg/kg) for 4 weeks.	Mouse	NMRI	②③④⑤⑦⑨
Peng, F.X., 2022 [47]	Apigenin	Flavones	oral	20 mg/kg/d	3 weeks	DHEA	DHEA s.c. (60 mg/kg) in sesame oil for 20 days.	Rat	SD	①②③⑦⑨
Liyanage, G.S., 2021 [31]	Soy isoflavone	Isoflavones	oral	50 mg/kg/d	3 weeks	Letrozole	Letrozole p.o. at 0.5 mg/kg (dissolved in 1% CMC) for 21 days.	Rat	SD	②③④⑤⑦⑧
Ma, X.H., 2021 [32]	Soy isoflavone	Isoflavones	oral	100 mg/kg/d	4 weeks	Letrozole	Letrozole at 1 mg/kg for 21 consecutive days.	Rat	SD	④⑤⑥⑦⑧
Zheng, S.Y., 2021 [39]	Quercetin	Flavonols	gavage	100 mg/kg/d	4 weeks	DHEA	DHEA s.c. at 6 mg/100 g dissolved in sesame oil for 20 days.	Rat	SD	②③④⑤⑥⑦⑧
Mihanfar, A., 2021 [40]	Quercetin	Flavonols	gavage	100 mg/kg/d	4 weeks	Letrozole	Letrozole (1 mg/kg) dissolved in CMC 0.5% for 21 consecutive days.	Rat	Wistar	⑦⑧⑨
Manzar, N., 2020 [33]	Soy isoflavone	Isoflavones	gavage	200 mg/kg/d	4 weeks	EV	EV injection at 4 mg/rat for 30 days.	Rat	Wistar	④⑤⑦⑧⑨
Hong, G., 2020 [35]	Catechin	Flavanones	oral	100 mg/kg/d	4 weeks	Insulin+hCG	Insulin s.c. started at 0.5 IU/day, increased by 0.5 IU per day, stopped in 6.0 IU/day, and 6.0 IU/day hCG s.c. twice a day	Mouse	C57BL/6	④⑤⑥⑧
Wang, W., 2019 [42]	Baicalin	Flavones	injection	50 mg/kg/d	4 weeks	DHEA	DHEA s.c. at 0.2 mL (6 mg/100 g) dissolved in sesame oil for 20 days.	Rat	Wistar	④⑤⑦⑧⑨
Yu, J., 2019 [43]	Baicalin	Flavones	oral	20 mg/kg/d	4 weeks	DHEA	DHEA s.c. at 0.2 mL (6 mg/100 g) dissolved in sesame oil for 20 days.	Rat	Wistar	④⑤⑥⑦⑧
Darabi, P., 2019 [46]	Apigenin	Flavones	gavage	40 mg/kg/d	3 weeks	EV	EV i.m. (4 mg/kg) dissolved in 0.2 mL sesame oil for 10 days.	Rat	Wistar	①②③④⑤⑥⑦⑧⑨
Jahan, S., 2018 [36]	Quercetin	Flavonols	gavage	30 mg/kg/d	3 weeks	Letrozole	Letrozole administered at 1 mg/kg dissolved in 0.5% CMC for 21 days.	Rat	SD	①②③⑦⑧⑨
Rajan, R.K., 2017 [34]	Soy isoflavone	Isoflavones	oral	100 mg/kg/d	2 weeks	Letrozole	Letrozole p.o. at 1 mg/kg for 21 days.	Rat	SD	⑦⑧
Hu, T., 2017 [45]	Rutin	Flavonols	gavage	100 mg/kg/d	3 weeks	DHEA	DHEA s.c. (6 mg/100g) dissolved in 0.2 mL of PBS for 20 consecutive days.	Rat	SD	④⑤⑥
Jahan, S., 2016 [44]	Rutin	Flavonols	oral	150 mg/kg/d	2 weeks	Letrozole	Letrozole p.o (1 mg/kg) dissolved in 0.5% CMC for 21 days.	Rat	SD	①②③⑦⑧⑨
Shah, K.N., 2015 [37]	Quercetin	Flavonols	oral	150 mg/kg/d	6 weeks	TP	TP s.c. (10 mg/kg) dissolved in olive oil daily for 6 weeks.	Rat	SD	③⑦

**Abbreviation:** CMC: carboxymethyl cellulose, DHEA: dehydroepiandrosterone, EV: estradiol valerate, hCG: human chorionic gonadotropin, i.m.: intramuscular injection, p.o.: peros, s.c.: subcutaneous injection, SD: Sprague Dawley, TE: testosterone enanthate, TP: testosterone propionate. **Outcomes:** ① number of atretic follicles, ② number of cystic follicles, ③ number of corpus luteum, ④ LH, ⑤ FSH, ⑥ LH/FSH, ⑦ free testosterone, ⑧ estradiol, ⑨ progesterone.

**Table 2 nutrients-14-04128-t002:** Subgroup analysis of flavonoids effect on count of cystic follicles.

Subgroups	N	Effect Sizes	95%CI		P-Heterogeneity
**Type of flavonoid**					<0.01 *
Soy isoflavone	1	0.12	−0.86	1.1	
Quercetin	3	−5.07	−9.7	−0.44	
Anthocyanin	1	−2.25	−3.31	−1.19	
Rutin	1	−1.98	−3.64	−0.33	
Apigenin	2	−4.68	−9.16	−0.2	
**Dose**					0.49
Low	5	−3.79	−7.23	−0.35	
Medium	2	−3.56	−6.36	−0.76	
High	1	−1.98	−3.64	−0.33	
**Duration of administration**					0.24
Short	2	−2.17	−3.07	−1.28	
Medium	6	−3.95	−6.76	−1.14	
**PCOS induction drug**					0.68
DHEA	3	−4.36	−7.73	−0.99	
TE	1	−2.25	−3.31	−1.19	
EV	1	−2.55	−3.96	−1.14	
Letrozole	3	−3.57	−9.21	2.07	

N: number of studies included, EV: estradiol valerate, TE: testosterone enanthate, * *p* < 0.05.

**Table 3 nutrients-14-04128-t003:** Subgroup analysis of flavonoids effect on count of corpus luteum.

Subgroups	N	Effect Sizes	95%CI		P-Heterogeneity
**Type of flavonoid**					0.03 *
Soy isoflavone	1	0.82	−0.22	1.85	
Quercetin	4	3.4	0.33	6.47	
Anthocyanin	1	0.78	−0.06	1.61	
Rutin	1	3.34	1.12	5.57	
Apigenin	2	2.34	1.18	3.5	
**Dose**					0.79
Low	5	2.19	1.24	3.14	
Medium	2	4.33	−2.9	11.57	
High	2	1.64	−1.32	4.6	
**Duration of administration**					0.53
Short	2	1.85	−0.63	4.32	
Medium	5	3.16	0.97	5.36	
Long	2	1.41	−0.95	3.77	
**PCOS induction drug**					0.02 *
DHEA	3	4.41	1.17	7.64	
TE	1	0.78	−0.06	1.61	
TP	1	0.3	−0.51	1.11	
EV	1	1.81	0.6	3.03	
Letrozole	3	2.29	0.51	4.07	

N: number of studies included, EV: estradiol valerate, TE: testosterone enanthate, TP: testosterone propionate, * *p* < 0.05.

**Table 4 nutrients-14-04128-t004:** Subgroup analysis of flavonoids effect on LH.

Subgroups	N	Effect Sizes	95%CI		P-Heterogeneity
**Type of flavonoid**					0.65
Soy isoflavone	3	−2.51	−6.07	1.05	
Quercetin	3	−3.39	−4.71	−2.07	
Anthocyanin	1	−2.06	−3.08	−1.04	
Rutin	1	−3.76	−6.18	−1.34	
Apigenin	1	−3.44	−5.12	−1.77	
Baicalin	2	−3.54	−9.34	2.25	
Catechin	1	−2.52	−3.75	−1.29	
**Dose**					<0.01 *
Low	5	−2.4	−4.63	−0.17	
Medium	5	−2.52	−3.13	−1.9	
High	2	−5.17	−6.54	−3.8	
**Duration of administration**					0.03 *
Short	1	−2.06	−3.08	−1.04	
Medium	10	−2.85	−4.13	−1.58	
Long	1	−4.75	−6.41	−3.08	
**PCOS induction drug**					0.03 *
DHEA	5	−3.13	−4.99	−1.27	
TE	1	−2.06	−3.08	−1.04	
TP	1	−4.75	−6.41	−3.08	
EV	2	−4.61	−7.16	−2.05	
Insulin + hCG	1	−2.52	−3.75	−1.29	
Letrozole	2	−0.83	−3.15	1.49	

N: number of studies included, EV: estradiol valerate, hCG: human chorionic gonadotropin, TE: testosterone enanthate, TP: testosterone propionate, * *p* < 0.05.

**Table 5 nutrients-14-04128-t005:** Subgroup analysis of flavonoids effect on LH/FSH.

Subgroups	N	Effect Sizes	95%CI		P-Heterogeneity
**Type of flavonoid**					0.3
Soy isoflavone	2	−2.59	−5.09	−0.09	
Quercetin	2	−6	−15.38	3.38	
Rutin	1	−4.64	−7.49	−1.79	
Apigenin	1	−1.24	−2.33	−0.14	
Baicalin	1	−1.73	−2.59	−0.88	
Catechin	1	−1.92	−3.01	−0.82	
**Dose**					0.08
Low	3	−1.51	−2.1	−0.92	
Medium	5	−4.31	−7.34	−1.28	
**Duration of administration**					0.17
Short	1	−1.54	−2.89	−0.18	
Medium	7	−3.31	−5.46	−1.16	
**PCOS induction drug**					0.4
DHEA	4	−4.38	−8.47	−0.29	
EV	1	−1.24	−2.33	−0.14	
Insulin + hCG	1	−1.92	−3.01	−0.82	
Letrozole	2	−2.59	−5.09	−0.09	

N: number of studies included, EV: estradiol valerate, hCG: human chorionic gonadotropin.

**Table 6 nutrients-14-04128-t006:** Subgroup analysis of flavonoids effect on free testosterone.

Subgroups	N	Effect Sizes	95%CI		P-Heterogeneity
**Type of flavonoid**					0.03 *
Soy isoflavone	5	−5.57	−10.17	−0.98	
Quercetin	5	−3.33	−4.70	−1.97	
Anthocyanin	1	−2.25	−3.31	−1.19	
Rutin	1	−0.56	−1.84	0.71	
Apigenin	2	−1.56	−2.35	−0.76	
Baicalin	2	−5.3	−14.06	3.46	
**Dose**					0.66
Low	6	−2.7	−4.84	−0.56	
Medium	7	−3.65	−5.97	−1.33	
High	3	−5.26	−11.04	0.53	
**Duration of administration**					0.44
Short	4	−4.35	−8.1	−2.51	
Medium	11	−3.28	−5.19	−1.36	
Long	1	−4.97	−6.69	−3.24	
**PCOS induction drug**					0.12
DHEA	5	−3.18	−5.91	−0.46	
TE	1	−2.25	−3.31	−1.19	
TP	1	−4.97	−6.69	−3.24	
EV	2	−6.06	−15.42	3.31	
Letrozole	7	−3.37	−5.93	−0.81	

N: number of studies included, EV: estradiol valerate, TE: testosterone enanthate, TP: testosterone propionate, * *p* < 0.05.

## Data Availability

Data generated from this systematic review are included in the manuscript and the Appendix A. Additional data are available upon request from the corresponding authors.

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
