# Peer review of "Efficacy of Flavonoids on Animal Models of Polycystic Ovary Syndrome: A Systematic Review and Meta-Analysis"

_nutrients, 2022, doi:10.3390/nu14194128_

Round 1
Reviewer 1 Report
The submitted manuscript is an interesting systematic review, which deals with efficacy of flavonoids on animal models of polycystic ovary syndrome. Since the polycystic ovary syndrome is a common-gynaecological disease, the topic of this manuscript is important.
The authors carried out comprehensive review and meta-analyses of data obtained in various refences. The used data and refences are relevant to the topic and aim of the paper. The employed data analyses are adequate and presented illustratively in many figures, especially in the file of supporting information. Additionally, the authors show in the article new aspects of the flavonoid effects on the polycystic ovary syndrome. Furthermore, the analysed data are properly discussed and support the conclusions.
Since I found no serious weaknesses in the manuscript, I recommend the paper for publication.
Author Response
Dear Reviewer 1,
Thank you very much for your reviewing and comments on our manuscript.
Reviewer 2 Report
The authors in this manuscript (nutrients-1933156) conducted a systematic review and meta-analysis of experimental studies to determine the efficacy of flavonoids on the pathology of PCOS on disease-relevant animal models. It is a comprehensive manuscript with well written English language. This analysis indicates the positive effects of flavonoids on ovarian histomorphology and hormonal status in animal models of PCOS whereas limitations of the study are also discussed.
Minor comments
· Figure 1: second box on the right column seems to be misplaced
· Please be consistent with the number of comparisons in 3.2. Numbers and words are alternating.
· Please check throughout the text for accuracy for example in the Discussion section, line 3-4 “inhibiting metal ions chelators”.
Author Response
Dear Reviewer 2,
Thank you very much for your reviewing and comments on our study, which is very helpful to us. After carefully reading your comments, we have responded to the suggestions point-by-point. All changes are made using Track Changes model so that they may be easily identified.
The authors in this manuscript (nutrients-1933156) conducted a systematic review and meta-analysis of experimental studies to determine the efficacy of flavonoids on the pathology of PCOS on disease-relevant animal models. It is a comprehensive manuscript with well written English language. This analysis indicates the positive effects of flavonoids on ovarian histomorphology and hormonal status in animal models of PCOS whereas limitations of the study are also discussed.
Minor comments
1. Figure 1: second box on the right column seems to be misplaced
Response: Thank you for pointing this out. We have moved the second box up to the right position (see New Figure 1).
2. Please be consistent with the number of comparisons in 3.2. Numbers and words are alternating.
Response: Thank you for the good suggestions. We have unified the numbers and words in the revised manuscript.
3. Please check throughout the text for accuracy for example in the Discussion section, line 3-4 “inhibiting metal ions chelators”.
Response: Thank you for pointing this out. We have checked the accuracy of the text throughout the new manuscript.